# Nutritional Conditions Modulate *C. neoformans* Extracellular Vesicles’ Capacity to Elicit Host Immune Response

**DOI:** 10.3390/microorganisms8111815

**Published:** 2020-11-18

**Authors:** Clara Luna Marina, Pedro Henrique Bürgel, Daniel Paiva Agostinho, Daniel Zamith-Miranda, Lucas de Oliveira Las-Casas, Aldo Henrique Tavares, Joshua Daniel Nosanchuk, Anamelia Lorenzetti Bocca

**Affiliations:** 1Laboratory of Applied Immunity, Institute of Biology Sciences, University of Brasília, Brasília, Distrito Federal 70910-900, Brazil; claraluna93@gmail.com (C.L.M.); phburgel@gmail.com (P.H.B.); lascasaslucas@gmail.com (L.d.O.L.-C.); 2Division of Infectious Diseases, Department of Medicine, Albert Einstein College of Medicine, New York, NY 10461, USA; daniel.molecular@gmail.com (D.P.A.); danielzamith@gmail.com (D.Z.-M.); josh.nosanchuk@einsteinmed.org (J.D.N.); 3Department of Microbiology and Immunology, Albert Einstein College of Medicine, Bronx, New York, NY 10461, USA; 4Faculty of Ceilândia, University of Brasilia, Distrito Federal 72220-275, Brazil; atavares@unb.br

**Keywords:** *Cryptococcus neoformans*, extracellular vesicles, inflammasome

## Abstract

*Cryptococcus neoformans* is a human pathogenic fungus that mainly afflicts immunocompromised patients. One of its virulence strategies is the production of extracellular vesicles (EVs), containing cargo with immunomodulatory properties. We evaluated EV’s characteristics produced by capsular and acapsular strains of *C. neoformans* (B3501 and ΔCap67, respectively) growing in nutritionally poor or rich media and co-cultures with bone marrow-derived macrophages or dendritic cells from C57BL/6 mice. EVs produced under a poor nutritional condition displayed a larger hydrodynamic size, contained more virulence compounds, and induced a more robust inflammatory pattern than those produced in a rich nutritional medium, independently of strain. We treated infected mice with EVs produced in the rich medium, and the EVs inhibited more genes related to the inflammasome than untreated infected mice. These findings suggest that the EVs participate in the pathogenic processes that result in the dissemination of *C. neoformans*. Thus, these results highlight the versatility of EVs’ properties during infection by *C. neoformans* in different tissues and support ongoing efforts to harness EVs to prevent and treat cryptococcosis.

## 1. Introduction

*Cryptococcus neoformans* is an environmental basidiomycetous yeast-like fungus. Mammals can inhale the spores or desiccated yeast, which transition to a yeast-like form and, if they avoid or subvert host immune responses, cause cryptococcosis. In humans, cryptococcosis is an opportunistic infection that affects patients with acquired immunodeficiency syndrome (AIDS) or other immunosuppressed conditions [1,2]. Infection in immunocompetent individuals typically results in an asymptomatic acquisition of the fungus or mild, self-limited pneumonia. In immunocompetent hosts, the fungi are usually eliminated or became latent, which can result in reactivation disease if the host immune system subsequently becomes compromised [3]. However, when *C. neoformans* infects individuals with defective cell-mediated immunity, especially those deficient in CD4^+^ T lymphocytes, the yeasts can migrate from the lung to cause disease in any organ, including the central nervous system (CNS) to cause a meningoencephalitis, which is uniformly lethal without antimicrobial therapy [4].

When the spores are inhaled and settled in the pulmonary alveoli, they differentiate into yeast and present several virulence strategies to survive and spread through host tissues and organs. One of *C. neoformans’* strategies is the presence of a complex polysaccharide capsule, which inhibits its recognition by the host immune system [5,6]. The fungus’ capsule allows it to survive inside macrophages and use these phagocytic cells as a “Trojan horse” to facilitate dissemination to tissues like the CNS [7]. The interaction of host opsonins and polysaccharides rearranges capsule molecules, which can be recognized by phagocytic cells [8]. Inside macrophages, the yeast cell regulates several characteristics, such as the host cell’s cytoskeleton and cell signaling organization to favor fungal survival [9]. Fungal survival and replication within macrophages have been associated with the activation status of the macrophage [10]. This is particularly curious as non-activated macrophages are incapable of killing the fungus [11]. However, macrophage activation is necessary for fungal survival and replication [10].

The first host immune response against *C. neoformans* and/or any other microorganisms is the innate immune response that recognizes foreign substances through microorganism pathogen-associated molecular patterns (PAMPs) [12]. This recognition is usually performed by innate immune cells like macrophages, neutrophils, and dendritic cells through specific receptors known as pattern recognition receptors (PRRs). These cells then present the foreign antigen to lymphocytes to develop a specific immune response [13]. In addition to membrane receptors, cytoplasmic receptors like nucleotide-binding oligomerization domain (NOD)-like receptors (NLR) recognize antigens present in the cell’s cytosol and form a multiprotein complex called the inflammasome, responsible for the cleavage of pro-caspase 1 followed by the production of the active forms of the pro-inflammatory interleukins (IL)-1β and IL-18 [14].

*C. neoformans* produces several virulence molecules related to its capacity to evade the host immune system and survive inside macrophages, like phospholipase B, acid phosphatase, laccase (melanin production), urease, and superoxide dismutase [15]. These molecules can be exported by the yeast cells directly or via the release of extracellular vesicles (EVs) [16,17]. The influence of EVs on the host immune response has been described with *C. neoformans* and other fungi species like *Candida albicans*, *Histoplasma capsulatum*, and *Paracoccidioides brasiliensis* [18,19,20,21]. Those studies show that the treatment of dendritic cells or macrophages with fungal EVs stimulates the production of pro-inflammatory cytokines and reactive oxygen and nitrogen species (ROS and RNS) [18,19]. The treatment of *C. neoformans* infected mice with yeast microvesicles enhances the infection, favoring fungal migration to the CNS and facilitating the crossing of the blood-brain barrier by yeast cells, resulting in higher fungal loads in the brain and accelerating the time to death of treated mice [22]. So far, cryptococcal EVs have only been studied in the context of growth in a carbon-source reduced medium to mimic similarities to some organs of the host environment.

During infection, *C. neoformans* needs to adapt to the host’s stressful conditions to survive and reproduce. The fungus must cross efficient physical barriers and overcome immunological pressures and the challenges of variable nutritional availability, gas concentration, and pH in different organs. The pulmonary alveoli and tissues, which are the initial site of infection, usually are regions with high oxygen concentrations (O_2_), but, after the consolidation of the infection, there is a significant local inflammatory response, leading to a decrease in mucosal oxygenation [23]. Pulmonary alveoli maintain fewer carbon and nitrogen sources as a strategy to avoid infection, turning it into a challenging environment for the fungus to develop [24,25,26]. However, other fluids and tissues have higher glucose and nitrate (NO_3_) concentration, such as the blood and brain [27,28]. Thus, *C. neoformans* has adapted responses for growth and survival under diverse stress conditions, including the thickening of their polysaccharide capsule and dynamic changes in EV secretion in the lung [29].

Considering the adaptation process in different nutritional conditions by *C. neoformans*, we analyzed the production of its EVs in different nutritional mediums from capsular and acapsular strain cultures and assessed the EVs for differences in morphology, cargo, and immunomodulatory capacity in vitro and in vivo. Their immunomodulatory factors were analyzed through interactions with bone marrow-derived macrophages and dendritic cells (BMDMs and BMDCs, respectively). Additionally, mice infected with *C. neoformans* were treated with EVs isolated from different strains to analyze genes alterations related to the mouse inflammatory response’s activation.

## 2. Materials and Methods 

### 2.1. Fungal Culture

In this study, a virulent capsular strain *C. neoformans* B3501 [30] and its hypocapsular avirulent mutant ΔCAP67 [31] were used to compare the influence of capsule molecules in the characteristics of the EVs. The fungi were stored at −80 °C in a freezing medium composed of 70% Sabouraud dextrose broth (SDB) (Kasvi—São José dos Pinhais, PR, Brazil) and 30% glycerol (Vetec—Duque de Caxias, RJ, Brazil). Before experiments, aliquots of the isolates were thawed and cultivated in Sabouraud dextrose agar (SDA) for 72 h, at 30 °C.

### 2.2. Vesicles Isolation

For isolation of *C. neoformans* EVs, one colony was inoculated in SDB for 48 h at 30 °C with shaking at 120 revolutions per minute (RPM). Then, the cells were counted and inoculated to a density of 1 × 10^5^ cells/mL in the different nutrient concentration media (minimal medium—MM (15 mM dextrose, 10 mM MgSO_4_, 29,4 mM KH_2_PO_4_, 13 mM glycine and 3 µM thiamine-HCl), SDB 50%, or SDB 100%, all adjusted to pH 5.5) and were incubated for 5 days at 30 °C with shaking at 120 RPM. The supernatant was filtered (0,45 µm), and the filtrate was concentrated using an ultrafiltration Amicon system with a 100 KDa membrane (Merck Millipore—Burlington, MA, USA). The concentrate was ultracentrifuged at 100,000× *g* for 1 h, and the pellet was suspended at 300 µL of phosphate-buffered saline (PBS) [19]. The *C. neoformans* EVs suspensions also contained glucuronoxylomannan (GXM), one of the main capsular fungal molecules. To purify the EVs and maximally eliminate capsular GXM, samples were subjected to a capture ELISA (enzyme-linked immunosorbent assay) using the antibody 18B7 with affinity by GXM (kindly provided by Arturo Casadevall, Johns Hopkins University, USA) [32].

### 2.3. Vesicles Characterization

EV cargoes were indirectly quantified by sterol and protein using the fluorimetric assays Amplex Red Cholesterol assay kit for ergosterol [19] quantification and the colorimetric assay Micro BCA Protein assay kit for protein quantification (both by Thermo Fisher Scientific—Waltham, MA, USA), following the manufacturer’s instructions. The EVs were characterized by dynamic light scattering (DLS) “Zetasizer Nano ZS” to analyze vesicles’ hydrodynamic diameters to observe its size and superficial characteristics.

### 2.4. Detection of Enzymatic Activity in Extracellular Vesicles (EVs)

Laccase, urease, and acid phosphatase were quantified in vesicle suspensions by spectrophotometric assays. The pellet obtained after 100,000× *g* ultracentrifugation was resuspended in PBS and serially diluted in specific media for each enzyme’s catalysis reactions. The laccase-specific reaction medium was composed of 0.2% (10 mM) l-3,4-dihydroxyphenylalanine (L-DOPA) in PBS, the medium for urease activity was 4% urea, 0.02% yeast extract, 0.002% phenol red, 0.273% KH_2_PO_4_ and 0.285% Na_2_HPO_4_. The reaction medium consisted of acetate buffer (pH 5) supplemented with 5 mg/mL p-nitrophenyl phosphate to determine acid phosphatase. Vesicle suspensions were incubated overnight at room temperature and protected from light. The reactions were quantified by reading at 450 nm (laccase), 405 nm (phosphatase), and 540 nm (urease) in a Spectramax spectrophotometer. The amount of EVs used was equivalent to 0.3 µg/mL of protein as the maximum concentration [17].

### 2.5. Mice and Ethical Agreement

All experimental procedures were carried out with C57BL/6 mice between 8 and 10 weeks of age. The work was approved by the Animal Ethics Committee of the University of Brasilia (UnBDoc number 55924/2016) and conducted according to the Brazilian Council for the Control of Animal Experimentation (CONCEA) guidelines.

### 2.6. Differentiation of Bone Marrow-Derived Macrophages (BMDM) and Bone Marrow-Derived Dendritic Cells (BMDC)

Bone marrow-derived macrophages (BMDM) and dendritic cells (BMDC) were obtained by mice femoral lavage, followed by erythrocyte lysis with a buffer containing ammonium chloride in Tris buffer and the resulting cells were counted in a hemocytometer. To obtain macrophages with a pro-inflammatory M1 profile and dendritic cells, 2 × 10^6^ cells were plated in 10 mL of Roswell Park Memorial Institute (RPMI)-1640 (Thermo Fisher Scientific) differentiation medium with gentamicin supplemented with 10% fetal bovine serum (FBS) and 20 ng/mL recombinant mouse granulocyte macrophage colony-stimulating factor GM-CSF (ImmunoTools—Friesoythe, Germany) and 50 µM β-mercaptoethanol (Sigma-Aldrich—St. Louis, MO, USA). After three days, an additional 10 mL of GM-CSF and β-mercaptoethanol differentiation medium was added. On the sixth day of differentiation, 10 mL of the culture medium volume was removed, centrifuged, and 10 mL of fresh medium containing GM-CSF and β-mercaptoethanol were added. On the eighth day of cultivation, dendritic cells (DCs) were considered to reside in the plaque supernatant, and macrophages were adherent. The supernatant containing the DCs was first removed, the cells obtained centrifuged, and the medium replaced with fresh RPMI-1640 medium containing 10% FBS. The adhered cells were detached by incubation with the Tryple Express dissociation solution (Life Technologies—Carlsbad, CA, USA) for 20 min at 37 °C. The cells were collected, and the solution was replaced by RPMI-1640 + 10% FBS; macrophages and DCs were counted and were ready for use [33]. Using flow cytometry (FACSVerse—BD Biosciences—San Jose, CA, USA), we determined that BMDC and BMDM cultured in GM-CSF were 87% positive for CD11c and MHC class II and 99,4% positive for CD11b, respectively.

### 2.7. Interaction of the EVs with BMDM and BMDC

BMDMs or BMDCs (1 × 10^6^ cells/mL) were incubated at 37 °C in a humidified 5% CO_2_ atmosphere. Cells were stimulated with lipopolysaccharide (LPS 500 ng/mL for 4 h—Sigma-Aldrich), providing the first signal for inflammasome activation. Additionally, cells were incubated for 18 h with or without EVs under different nutritional conditions (40 ng/mL—sterol content). After that, cells were treated with nigericin (20 µM for 40 min—Invivogen—San Diego, CA, USA), providing the second signal for inflammasome activation. Controls included conditions without LPS or nigericin. After interactions, supernatants were collected for cytokine quantification by ELISA assay.

### 2.8. Cytokine Quantification

The cell-free supernatants of the BMDM and BMDC cultures were harvested for measurements of interleukins IL-1β, IL-6, and tumor necrosis factor (TNF)-α (Ready-Set-Go! Kit—Thermo Fisher Scientific) concentrations using ELISA. The data were expressed as pg/mL ± the standard deviation (SD) of two to three independent experiments conducted in triplicate.

### 2.9. Murine Infection and Treatment with EVs

The influence of EV treatment on the host immune response was analyzed using groups of C57Bl/6 wild-type mice infected intratracheally with the virulent *C. neoformans* strain B3501 (2 × 10^4^ cryptococcal cells/animal) followed by treatment with 20 µL of intranasal PBS or EVs from B3501 or ΔCap67 EVs (50 µg of protein content) produced in rich medium (SDB 100%) [22]. Before the surgery, animals were anesthetized by Ketamine (Vetnil—Louveira, SP, Brazil) and Xylazine (Ceva—Paulínia, SP, Brazil) (1:1). One group of animals received treatment on days one and three after infection and were euthanized 5 days after infection. The other group received treatment two and four days after infection and were euthanized 15 days post-infection (dpi). The lungs and brains were surgically removed to analyze the fungal burden, histopathology, cytokine secretion, and messenger ribonucleic acid (mRNA) transcript levels. To quantify the fungal burden, the tissues obtained were weighed and macerated in 1 mL of PBS using a glass macerator. The homogenized tissues were plated onto SDA plates, incubated at 30 °C for 48 h, and then colony forming units (CFUs) were determined. The remaining homogenate was centrifuged, and the supernatant was collected for cytokine quantitation by ELISA. For histopathological analysis, small organ fragments were fixed in 10% formaldehyde.

### 2.10. RNA Extraction and Reverse-Transcription Polymerase Chain Reaction (RT-PCR)

For RNA extraction, lung fragments were macerated using 1 mL of the QIAzol Lysis Reagent buffer (Qiagen—Hilden, Germany). The subsequent steps were done using the RNeasy mini-kit (Qiagen), following the manufacturer’s instructions. The extracted RNA was quantified using the Implen nanophotometer, and an electrophoresis gel was made for certification of RNA integrity. The intact RNA was converted to a complementary deoxyribonucleic acid (cDNA) using the RT^2^ First Strand Kit (Qiagen), following the manufacturer’s recommendations. The cDNA produced was used to detect differentially expressed genes in the lungs using the kit “Mouse Inflammasomes RT2 Profiler PCR Array” (Qiagen), which recognizes 84 essential genes involved in inflammasome activation and NOD signaling (NLR). The data analysis was done in Qiagen online software called “GeneGlobe Data Analysis Center”.

The constituent genes and other controls were: microglobulin Beta-2 (*BM2*), glyceraldehyde-3-phosphate dehydrogenase (*Gapdh*), glucuronidase beta (*Gusb*), 90 alpha (cytosolic) heat shock protein, class B1 member (*Hsp90ab1*), Mouse Genomic DNA Contamination (*MGDC*), reverse transcription control (*RTC*), positive PCR control (*PPC*). Normalization was undertaken manually, and the control genes used were *Hsp90ab1* and *Gusb*. The fold change used in all analyses was 2X, and the minimum detection Cycle Threshold (CT) was considered from 30.

### 2.11. Statistical Analysis 

The results presented in this work refer to a representative triplicate group experiment of at least two independent assays. Statistical analysis was conducted using GraphPad Prism v. 7.0 software, using one-way analysis of variance (ANOVA) or student’s *t*-test methods to compare all results. *p*-Values of less than 0.05 were considered significant, shown with *, when *p* < 0.05; with **, when *p* < 0.01; with *** when *p* < 0.001; with **** when *p* < 0.0001 and with ns when *p* > 0.05. Error bars are standard error of the mean (SEM). For normalization, the data were median centered and log-transformed.

## 3. Results

### 3.1. The Nutrient Complexity of the Medium Modulates the EVs’ Hydrodynamic Diameter and Virulence Factors

We investigated the size of EVs produced by the virulent strain B3501 and its hypocapsular and avirulent, ΔCap67, in both rich and poor media (Figure 1). Dynamic light scattering (DLS) experiments revealed that both strains produced vesicles of different sizes when grown in different media. Both strains secreted larger EVs when grown in the poor medium compared to EVs produced in the rich medium (Figure 1A–E). Moreover, B3501 EVs presented a homogeneous size pattern, represented by one peak, while ΔCap67 EVs presented a heterogeneous pattern, characterized by two peaks (Figure 1B,D).

EVs transport proteins to the extracellular milieu, such as laccases and ureases, which have a role in fungal virulence. Ergosterol is the main component of the EV membrane. To understand if the increase in the EVs’ size affects their capacity to transport these proteins, we analyzed the protein/sterol ratio (Figure 1F). The ratio of protein/sterol increased for both strains as the vesicle size increased, showing that the EV not only became larger, but they also contained higher amounts of proteins (Figure 1F).

*C. neoformans* EVs contain GXM as well as virulence associated enzymes, like laccase and urease [16]. We investigated the quantity of GXM and activity for laccase and urease in the EVs isolated from the different isolates under different growth conditions. A significant increase in GXM was found in EVs from B3501 and ΔCap67 grown in MM compared to SDB (Figure 1G). Laccase activity increased in EVs from B3501 grown in MM compared to SDB, while in ΔCap67 EVs, laccase activity was lower in EVs grown in MM (Figure 1H). Urease activity increased in EVs from MM in both strains (Figure 1I).

### 3.2. EVs Produced in a Rich Medium Negatively Modulate the Inflammatory Activation of BMDCs and BMDMs 

To verify EVs’ role in inflammasome activation, BMDMs and BMDCs were co-cultured with 40 ng/mL of EV sterol content from both serotypes produced in poor or rich media (SDB 50% or 100% respectively). These co-cultures were either in the presence of LPS (a known first signal), nigericin (a second signal inducer), or both (Figure 2). 

B3501 EVs, without other stimuli, were not able to stimulate the production of IL-1β by BMDCs, although EVs produced at SDB 50%, but not at SDB 100%, stimulated the production of TNF-α (Figure 2A). When EVs were added with nigericin, EVs produced in SDB 50% induced the production of high amounts of IL-1β by BMDCs, while EVs produced at SDB 100% did not, indicating that the B3501 SDB 50% EVs acted as the first signal for inflammasome activation, whereas the EVs produced in the rich media were unable to function in this manner (Figure 2A). A similar pattern in cytokine production was also observed for TNF-α levels. ΔCap67 EVs were also unable to stimulate IL-1β or TNF-α by BMDCs when inoculated alone with the cells or only with LPS or nigericin (Figure 2B). The results showed in Figure 2A and B indicate that B3501 and ΔCap67 EVs, produced in a rich medium, did not promote an inflammatory response by DCs, considering that they didn’t stimulate the production of IL-1β and TNF-α, which probably is not inflammasome pathway-dependent, since TNF-α is a cytokine that does not depend on this pathway for its production.

The response to stimulation of BMDMs with EVs (Figure 2C–D) was different from that observed with the BMDCs. B3501 EVs produced in SDB 50%, but not in SDB 100% stimulated IL-1β production by BMDMs when incubated only with LPS. The production was significantly higher when incubated only with nigericin (Figure 2C), showing that these EVs also have characteristics of the first signal for macrophage inflammasome activation. BMDMs incubated with ΔCap67 EVs (Figure 2D), and nigericin led to higher production of IL-1β than when incubated with LPS (Figure 2D). TNF-α production was stimulated by all EVs when incubated alone with BMDM, indicating that the stimulation observed is probably not dependent on the inflammasome pathway. These data together indicate that B3501 and ΔCap67 EVs can function as the first signal for the activation of BMDM’s inflammasome, stimulating the production of high amounts of IL-1β, when nigericin in added together, and EVs produced in a poor medium (SDB 50%) tended to be more robust in boosting these cytokines. In comparison, EVs produced in a rich medium (SDB 100%) tended to inhibit their production. This modulation is not only dependent on the inflammasome pathway since TNF-α production is also affected.

### 3.3. EVs from Rich Medium Induce an Anti-Inflammatory Response in Infected Mice Lungs at 15 Days Post-Infection with an Increase in the Fungal Burden

To find out whether EVs produced in the rich medium (SDB 100%) can also inhibit inflammatory pathways in vivo, we carried out two independent protocols, where C57BL/6 mice infected with *C. neoformans* B3501 received intranasal treatment with B3501 or ΔCap67 EVs produced in SDB 100%. EV treatments were administered at the beginning of the infection. In the first protocol, mice were treated with EV at 1- and 3-days post-infection (dpi), and then the mice were euthanized 5 dpi. In the second, the EV treatment occurred at 2 and 4 dpi, and the mice were euthanized 15 dpi. After euthanizing the animals, lungs and brains were collected for histopathological analysis, RNA extraction, and fungal burden. The fungal load of EVs in treated mice was lower than in non-treated mice at 5 dpi (Figure 3A); however, the fungal load in the lungs of treated mice with 15 dpi was higher than non-treated mice (Figure 3B). This result indicates a variable influence of EV treatment in the course of the disease, wherein the beginning of infection leads to increased recognition of fungi due to the presence of EVs and the elimination of yeast cells. However, the 15-day data demonstrates that EVs enhance disease. In contrast, the untreated mice were found, as expected with the low (2 × 10^4^/animal) infecting dose, to be controlling the infection as the CFUs in the untreated mice were significantly lower at day 15 compared to day 5 (2.99 vs. 4.64 CFU, *p* < 0.05), albeit these comparisons are between two different experiments.

To understand if cytokine production was related to this CFU reduction at 15 dpi, we quantified IL-1β, TNF-α, and IL-6 production in lungs 5 dpi (Figure 3C,D) and lungs and brains 15 dpi (Figure 3E–J). With 5 days of infection, the cytokine levels in the lungs of EV-treated mice were similar to those of untreated-mice (Figure 3C,D). In contrast, at 15 dpi, there were significant reductions of IL-1β and TNF-α, and a trend to lower amounts of IL-6 in the lungs of treated mice with EVs from either *C. neoformans* isolate (Figure 3E–G). In the brain, there were reductions of IL-1β, TNF-α, and IL-6 in mice treated with B3501 EVs (Figure 3H–J), while ΔCap67 EVs only decreased IL-1β production. These results could reflect the capacity of the virulent strain to infect the CNS. These results are compatible with CFU results (Figure 3A,B), in which the reduction in pro-inflammatory cytokine production at a late stage of infection favors fungal survival and growth. There was no fungi growth from the brain, probably due to the low inoculum used in these experiments and the time analyzed.

Histopathological analysis showed a decrease in infection, inflammation, and cell migration in lungs treated with EVs, mainly ΔCap67 EVs after 5 days of infection, while there were no differences in histopathological analysis after 15 dpi (Appendix A). This data is concordant with CFU results (Figure 3A,B), in which there is a reduction of fungal load in the lungs at 15 dpi compared with 5 dpi.

### 3.4. Rich-Medium EVs Negatively Modulate Inflammasome Gene Expression in the Lungs of Mice Infected with C. Neoformans after 5 and 15 Days Post-Infection (dpi)

To understand the mechanism of the reduction in cytokine levels, we analyzed the gene expression in the lungs of infected mice using the kit Mouse Inflammasomes RT^2^ Profiler PCR Array, which compares 84 genes related to inflammasome differentially expressed in mice treated or not with *C. neoformans* EVs. At 5 dpi, there was a significant reduction of mRNA amounts in mice treated with B3501 EVs compared with untreated infected mice and a less significant reduction in mice treated with ΔCap67 EVs (Figure 4A,B). The genes most downregulated in the lungs of B3501 EV-treated mice were *Bcl2l1* (Bcl2-like protein 1), *Mapk1* (Mitogen-activated protein kinase 1), *Mapk13* (Mitogen-activated protein kinase 13), *Birc2* (regulates caspases and apoptosis), *Il18* (Interleukin-18) and *Card6* (Caspase recruitment domain family, member 6), genes related to NF-ϰB and caspases signaling and MAP kinase signal transduction pathway. Other important genes that were downregulated were *CD40* ligand, caspases 1 and 8, the gene precursors of adapter protein Myd 88, intracellular Nod-like receptors NLRP3, NLRC4, NLRP5, and of cytokines IL-1β, IL-33, IL-12, and TNF. The genes significantly reduced in both treatments were *Aim2* (Absent in melanoma 2), *Bcl2*, *Ccl7* (Chemokine ligand 7), *Cflar* (CASP8 and FADD-like apoptosis regulator), *Chuk* (Conserved helix-loop-helix ubiquitous kinase), *Ikbkg* (Inhibitor of kappaB kinase gamma), *Ripk2* (Receptor TNFRSF-interacting serine-threonine kinase 2), *Mok* (Serine/threonine kinase 30) and *Sugt1* (Serine/threonine kinase 30) (Figure 4A,B,F).

After 15 dpi, although to a lesser extent than found with B3501 EV at 5 dpi, there were significant downregulations of inflammasome genes in the lung of both EVs treated mice, most of them similar to the downregulation in mice euthanized at 5 dpi, but, in this case, most genes were downregulated by both treatments. Of particular note are *Bcl2*, *Birc2*, *Card6*, *Ikbkg*, the chemokine genes *Cxcl1* and *Cxcl3* (chemokine ligand 1 and 3) and cytokine genes *Ifng* (Interferon-gamma), *Ifnb1* (Interferon-beta1), as well as the precursors of interleukins IL12, IL18, Il1β, and TNF and of adapter protein Myd88 and Noll-Like Receptors (Figure 4C,D,G). These data together indicate that in the early period of infection, EV treatment favors the recognition of *C. neoformans*, promoting cytokine production and control of fungal growth, explaining the low CFU in treated groups at 5 dpi. However, at 15 dpi, EV treatment also inhibited many genes related to inflammasome activation, as shown in the transcript cytokine production at 15 dpi. A network analysis of interacting inflammasome genes is shown in Figure 4E. It demonstrates how the genes downregulated with the treatment with EVs are related, especially how the regulation of intracellular receptors like NLRs and other intermediate proteins like Myd88 influence the production of pro-inflammatory cytokines such as IL18, Il1β, and TNF.

## 4. Discussion

EVs have been well described as a strategy used by several prokaryotic and eukaryotic pathogens for exporting compounds that can facilitate their survival in a host organism [34,35]. *C. neoformans* EVs are composed of lipids, proteins, polysaccharides, and pigmented structures with important functions for the biology of the fungus. Among these host-related functions are the evasion of the immune system, modulation of macrophages’ phagocytic activity, and cytokine secretion by macrophages and dendritic cells [36]. In this work, we analyzed the structural and immunomodulatory differences between EVs from an encapsulated *C. neoformans* strain and its acapsular mutant and defined differences between EVs produced in mediums with high or low nutritional availability. *C. neoformans* can modulate the production and cargo of EVs [37], suggesting dynamic changes in this secretion mechanism according to the environment. This is not unique to *C. neoformans* as *Histoplasma capsulatum* dynamically alters its loading and release of EV in response to available nutrients [38].

In vivo infection, the presence of mammalian serum, or other stressful situations, such as nutritional limitation, CO_2_, and iron limitation, enhance diverse virulence responses by *C. neoformans*. One example is the thickening of the polysaccharide capsule and production of titan cells, which hinder the host immune system’s recognition of fungi, favoring a non-protective host response [29]. Similarly, *A. fumigatus*, in stress conditions such as hypoxia, changes its morphotype, increasing host inflammation, disease progression, and murine mortality [39]. We now show that a stressful microenvironment with low nutrient availability results in *C. neoformans* producing EVs that are larger than those produced in a rich medium, in addition to exhibiting a higher content of virulence factors such as GXM, acid phosphatase, urease, laccase, and heat shock proteins. These molecules have previously been described in *C. neoformans* EVs produced in a poor medium [16]. The same group has previously shown that EVs from the encapsulated strain were higher in hydrodynamic diameter than those from an acapsular mutant [40], similar to our data.

The production of EVs as a strategy for the liberation of virulence compounds has also been described in Gram-negative and recently Gram-positive bacteria, containing virulence factors, compounds related to antibiotic resistance, and host-parasite interaction [39]. Growth conditions, such as temperature and nutrient availability in the culture medium, directly affect the quantity and characteristics of EVs produced by Gram-negative bacteria, varying from species to species. Simple increases in temperature during growth can result in an increased production of EVs. This favors the proliferation of microorganisms, keeping them in the log phase of growth and consequently supporting EVs production. Regarding the availability of nutrients, *Lysobacter* sp. secreted more EVs when subjected to low availability of nutrients, while *Pseudomonas fragi* produced less under these conditions [41]. In our work with *C. neoformans*, the medium rich in nutrients stimulated higher EV production, although the EVs were smaller and had a lower protein content than those produced in the nutrient restricted environment.

EVs from encapsulated (HEC3393 and B3501) and acapsular (ΔCap 67) isotypes of *C. neoformans* are actively incorporated by macrophages, modulating their functions without generating acute toxicity to these cells. The EVs of both serotypes of *C. neoformans* were able to modulate nitric oxide production and cytokine levels [19]. However, EVs from the acapsular mutant induced higher production of nitric oxide and TNF-α by macrophages and lower production of TGF-β and IL-10 than the EVs from the encapsulated strain. Also, when these macrophages pretreated with secreted EVs were incubated with encapsulated and non-opsonized *C. neoformans*, the vesicular content activated phagocytosis and macrophage microbicidal activities, enhancing the fungicidal activity of the host cells. This activation was more efficiently induced by acapsular EVs, which can be explained by the absence of GXM in these EVs [16,19]. Our results present a similar pattern, where ΔCap67 EVs stimulated higher production of IL-1β by macrophages than those from B3501. Previous studies demonstrated that *C. neoformans* and *C. albicans* EVs produced in a nutritionally deficient environment (minimal medium) were able to stimulate the immune system in general [18,19]. Our results corroborate this, where EVs produced in the deficient medium stimulated the inflammasome by macrophages more than those produced in the rich medium, and these EVs appeared to act as a first signal for the inflammasome activation.

The production of virulence molecules with immunomodulatory functions has been described in other species of fungi, such as *H. capsulatum* [20], *P. brasiliensis* [21], and *C. albicans* [18], being secreted freely or associated with EVs. In *H. capsulatum*, *P. brasiliensis*, and *C. albicans*, the secreted EVs have antigenic epitopes recognized in the serum of infected patients, indicating a potential influence of EVs on the adaptive response against these infections. The sensitivity of yeast cells to dynamic changes in EV biology is demonstrated by antibody binding to *H. capsulatum*, significantly modifying the characteristics of its EVs in size, abundance, and content in a dose-dependent manner [42,43]. In *P. brasiliensis*, EVs activated an inflammatory profile in macrophages, promoting the secretion of IL-1β, TNF-α, IL-6, and IL-12p70 and inducing macrophages to change from an M2 anti-inflammatory profile to an M1 type inflammatory profile [44]. Unlike the EVs of *P. brasiliensis*, those of *C. albicans* induce a less inflammatory profile, inducing the secretion of the anti-inflammatory cytokines TGF-β and IL-10 by dendritic cells and macrophages, respectively, and of TNF-α by both. This profile makes sense since *C. albicans* is a commensal organism, and its presence does not stimulate an exacerbated immune response [18,35]. 

The modulation of cytokine production in vitro by macrophages and dendritic cells or in vivo by mice treated with EVs from *C. neoformans* produced in different nutritional culture mediums may indicate an adaptation of *C. neoformans* to survive in different host organs and regions. Inflammatory sites are regions with a lower availability of nutrients, higher CO_2_ concentration, and lower O_2_ concentration than the external environment [22]. To survive in such conditions, fungal species like *C. albicans* and *C. neoformans* induce the expression of genes related to alternative processes for obtaining carbon, which removes this nutrient from low carbon sources, such as acetate and fatty acids. The main pathway activated by these pathogens is the glyoxylate cycle, which allows the synthesis of glucose and Krebs Cycle intermediates from Acetyl-CoA, and the main inducing molecule of this cycle is isocitrate lyase (ICL). This molecule is also overexpressed in nutrient-poor phagosomes in immune cells and inhibited in nutrient-rich regions, such as blood [23].

During lung homeostasis, there is high availability of nutrients, mainly due to the secretion of mucus rich in nitrogenous compounds and glucose, responsible for humidifying the airways and preventing the entrance of foreign microorganisms. However, this organ’s epithelium is composed of amino acid transporters responsible for keeping the concentration of nutrients in the lumen low, making it difficult for pathogenic microorganisms to survive [45]. Besides that, when a pathogen such as *C. neoformans* manages to enter the tissue, there is a significant activation of the inflammatory response, inducing phagocytosis of microorganisms by the resident or newly migrated macrophages. The inflammatory environment and the phagolysosomes’ interior have even lower nutrient availability [24,25].

Our data show that EVs produced in a nutrient-poor environment tend to stimulate the inflammatory response, and this probably occurs in the initial context of infection, which favors yeast phagocytosis. This stimulus is favorable to *C. neoformans* since this species survives well inside phagocytes and uses them to spread to other organs, including the CNS. *C. neoformans* crosses the blood-brain barrier (BBB) through different mechanisms. Although the yeast can enter the CNS within phagocytes, the fungus can also be released (e.g., by host cell lysis or non-lytic exocytosis) [46] near the BBB where it can interact with CD44 receptors, which facilitates the fungus’ entrance [47]. IL-6 levels are related to the integrity of the blood-brain barrier. IL-6^-/-^ mice infected with *C. neoformans* showed a higher permeability of the blood-brain barrier, followed by a more significant cerebral fungal load and higher mortality than wild-type mice for IL-6 [48]. Hence, the lower levels of IL-6 in the brains of B3501 EV treated mice may be associated with fewer yeast cells traversing the BBB. 

For *C. neoformans*, brain tissue is less stressful and has a high availability of glucose and other nutrients than the lung’s inflammatory environment. This is physiologically expected as the proper functioning of nerve cells depends on the high availability of such nutrients [23,26]. We believe that, upon reaching the CNS, high concentrations of glucose and nitrogen induce *C. neoformans* to produce EVs with an inhibitory function of the immune system, as shown here in both in vitro and in vivo models. This change in the characteristics of EV produced in response to nutritional resources is interesting since there is remarkably little inflammation in the brain during cryptococcosis, which is beneficial to *C. neoformans*. Corroborating this hypothesis, intravenous EV treatment increased the crossing of the BBB by *C. neoformans* and the cerebral fungal load, which reinforces the importance of EVs in the process of migration and colonization of the fungus in brain tissue [21].

## 5. Conclusions

The remarkable importance of EVs during cryptococcosis continues to be characterized, and we show that EVs are associated with the migration of the fungus from the lungs to the CNS, which is linked to alterations in the pattern of inflammasome activation and other inflammatory processes in both tissues. In our model, the production and characteristics of EVs are related to nutrient availability, which mimic the different conditions in the tissue in which the fungus grows. EVs can dynamically modulate the host response, influencing cell migration and inflammatory response, in a manner favorable to the fungus or the host, depending on the source and quantity of EVs. However, further studies are necessary to show how this occurs and identify the mechanisms that cause such inhibition. A greater understanding of the characteristics and functions of *C. neoformans* EVs is essential for understanding the biology of this species, which will facilitate the development of new approaches to combat cryptococcosis.

## Figures and Tables

**Figure 1 microorganisms-08-01815-f001:**
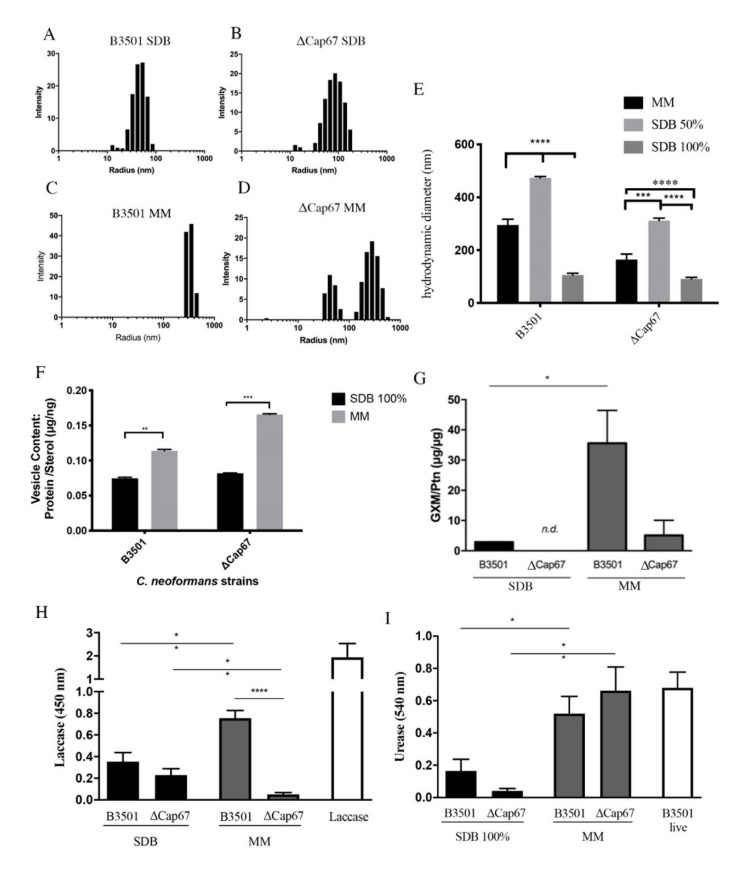
Extracellular vesicles (EVs) produced in a nutrient-poor medium have a higher hydrodynamic diameter and different virulence molecules activities than those produced in a nutrient-rich medium. EVs were isolated from *C. neoformans* strains B3501 (**A**,**C**) and ΔCap67 (**B**,**D**) grown in a poor medium (minimal medium (MM)—(**C**,**D**) or Sabouraud dextrose broth (SDB) 50%—(**E**) or in a rich medium (SDB 100%) (**A**,**B**,**E**) and were submitted to dynamic light scattering (DLS) analysis. Subfigures (**A**–**D**) show the distribution of EVs in suspension according to the average radius in nanometers (nm), and (**E**) compares hydrodynamic diameter averages depending on the culture medium utilized. Protein/sterol ratio of EVs was quantified using colorimetric assays Amplex Red Cholesterol assay kit for sterol quantification and Micro BCA Protein assay kit for protein quantification (**F**); glucuronoxylomannan (GXM) content was measured by enzyme-linked immunosorbent assay (ELISA) (**G**). Laccase (**H**) and urease (**I**) enzymatic activities were measured spectrophotometrically. Laccase and B3501 live are positive controls for the enzymatic activity. For all plots, * indicates *p* < 0.05, ** indicates *p* < 0.01, *** indicates *p* < 0.001, ****** indicates *p* < 0.0001 according to one-way analysis of variance (ANOVA) (**E**–**F**) and Student’s *t*-test (fig. **G**–**I**). *n.d.* means non detected. The data are expressed as the mean ± standard error of the mean (SEM) of triplicates and represent at least two (**G**–**I**) or three (**A**–**F**) independent experiments.

**Figure 2 microorganisms-08-01815-f002:**
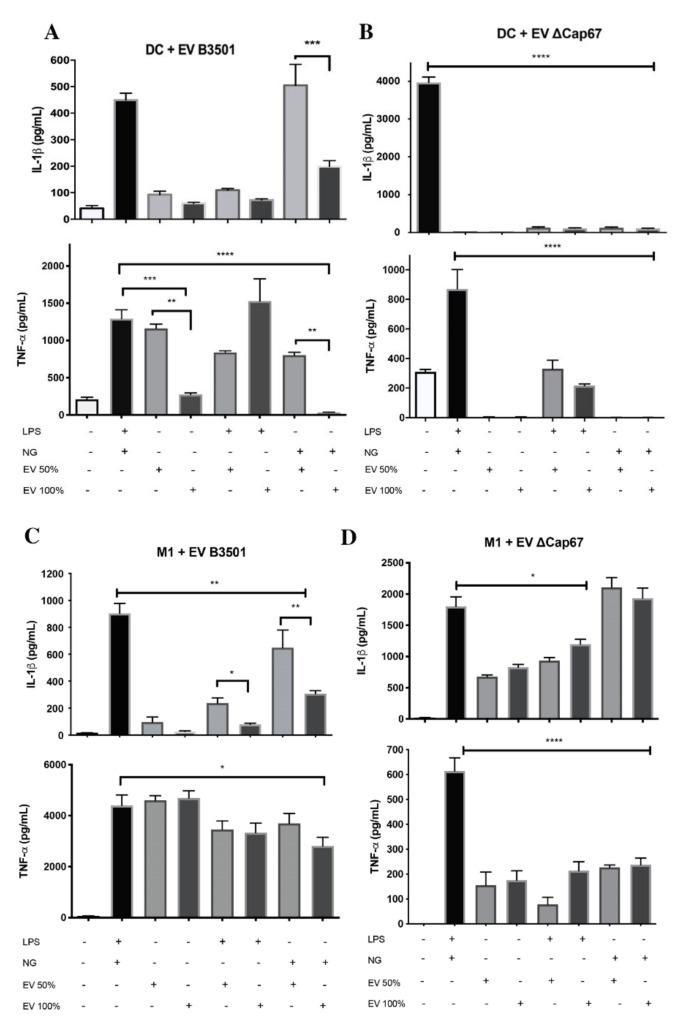
Extracellular vesicles modulate the production of inflammatory cytokines by bone marrow-derived dendritic cells (BMDCs) and bone marrow-derived macrophages (BMDMs). EVs (40 ng/mL of sterol content) isolated from *C. neoformans* strains B3501 (**A**,**C**) and ΔCap67 (**B**,**D**) grown in SDB 50% or in SDB 100% were incubated with 1 × 10^6^ cells/mL of BMDCs (**A**,**B**) or BMDMs (**C**,**D**) in the presence or not of lipopolysaccharide (LPS, 500 ng/mL) and/or nigericin (20 µM) for 24 h at 37 °C and 5% CO_2_ followed by determinations of cytokine production by enzyme-linked immunosorbent assay (ELISA). * indicates significant difference *p* < 0.05, ** indicates *p* < 0.01, *** indicates *p* < 0.001, **** indicates *p* < 0.0001 according to one-way ANOVA. The data are expressed as the mean ± SEM of triplicates and represent at least three independent experiments.

**Figure 3 microorganisms-08-01815-f003:**
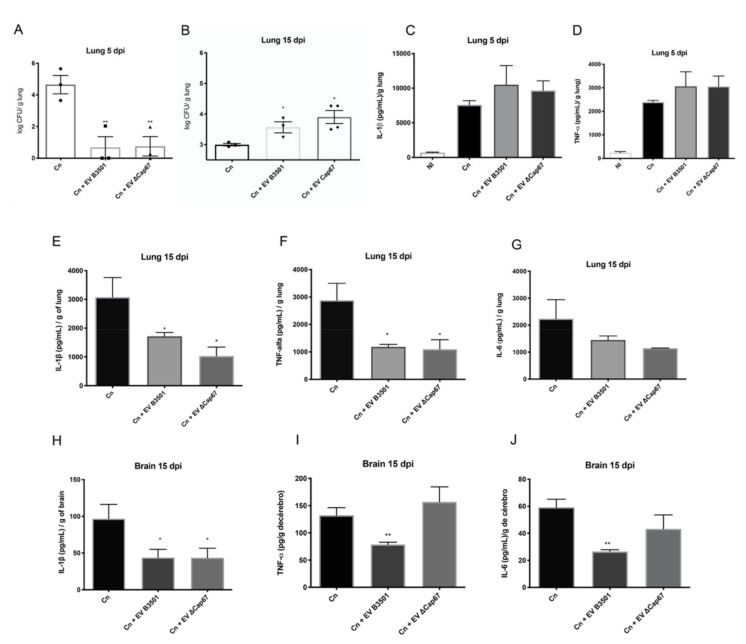
Treatment with EVs produced in a rich medium induces an anti-inflammatory response in the lungs of infected mice after 15 days of infection with an increase of the fungal load in the lungs. C57Bl/6 mice were infected with *C. neoformans* (2 × 10^4^ cells/animal) intratracheally and treated intranasally with 50 µg of EVs (protein content) produced in SDB 100%. Mice were treated one and three days post-infection (dpi) and euthanized 5 dpi (**A**,**C**,**D**) or treated two and four dpi and euthanized 15 dpi (**B**,**E**–**J**). Lungs were macerated and spread on Sabouraud dextrose agar (SDA) plates and were cultivated for 3 days at 30 °C to quantify colony forming units (CFUs) (**A**,**B**). The macerated supernatants were quantified for cytokine content with ELISA assay (**C**–**J**). * indicates *p* < 0.05, ** indicates *p* < 0.01 (significant difference comparing with Cn group), according to one-way ANOVA. The data are expressed as the mean ± SEM of triplicates and represent at least three independent experiments.

**Figure 4 microorganisms-08-01815-f004:**
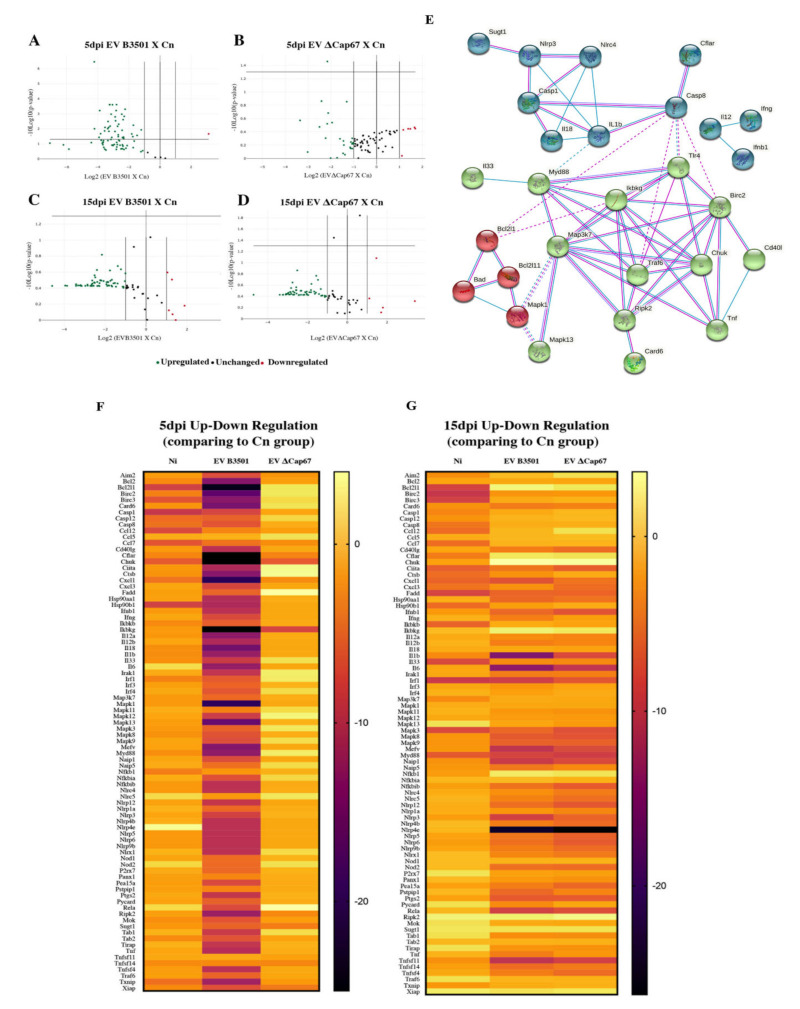
Treatment with EVs produced in a rich medium negatively modulated gene expression related to inflammasome activation in mice infected with *C. neoformans* at 5 and 15 dpi. C57Bl/6 mice were infected with *C. neoformans* (2 × 10^4^ cells/animal) intratracheally and treated intranasally with 50 µg of EVs (protein content) produced in SDB 100%. Mice were treated one and three dpi and euthanized 5 dpi (**A**,**B**,**F**) or treated two and four dpi and euthanized 15 dpi (**C**,**D**,**G**). Lungs of the infected and treated mice were macerated with QIAzol Lysis Reagent buffer for RNA extraction using RNeasy mini-kit (Qiagen). RNA was converted to cDNA using the RT^2^ First Strand Kit (Qiagen), which was used to detect differentially expressed genes in the lungs using the kit “Mouse Inflammasomes RT^2^ Profiler PCR Array” (Qiagen). The Volcanos Plot compared gene expression in the treated group with untreated groups and identified significant gene expression changes by plotting the log2 of the fold changes in gene expression on the x-axis versus their statistical significance on the y-axis. The center vertical line indicates unchanged gene expression, while the two outer vertical lines indicate the selected fold regulation threshold. The horizontal line indicates the *p*-value threshold (1,3). Genes with data points in the far upper left (down-regulated) and far upper right (up-regulated) sections meet the selected fold regulation and *p*-value thresholds (**A**–**D**). Network analysis of interacting inflammasome genes separated in color clusters of more related genes produced using String online software (**E**). The heatmaps show up and down-regulation on gene expression in the lungs of treated mice compared with untreated (**F**–**G**). The experiment was performed with biological triplicates and are representative of at least three independent experiments.

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
