# Peer review of "Nutritional Conditions Modulate C. neoformans Extracellular Vesicles’ Capacity to Elicit Host Immune Response"

_microorganisms, 2020, doi:10.3390/microorganisms8111815_

Round 1

Reviewer 1 Report

In this study the authors analyzed the production of EVs isolated from Cryptococcus that was grown in different nutritional conditions and from capsular and acap strains. The EVs were assessed for differences in morphology, cargo, and immunomodulatory capacity in vitro and in vivo. The study was predicated on previous published articles that had shown that EVs contribute to the virulence of Cryptococcus but the cryptococcal EVs studied previously were only studied in the context of growth in a carbon-source reduced medium to mimic the host environment. In contrast, the results discussed here were obtained from EV isolated from Cryptococcus grown in 50% SBD (Sabouraud Dextrose media) or 100% SD media or in Minimal media.

Although the manuscript describes some interesting data, there are several aspects of the study/experiments that are not addressed (please see the concerns below) and this makes the interpretation of the results confusing and open-ended. Also, given previous published studies that have shown similar results to the ones shown here, there is very little new information provided by the current study other than the EVs from Cryptococcus grown in rich media or minimal media may behave differently. But given that published articles have described EVs from low carbon sources mimicking the host environment, it is not clear how the EVs isolated here from rich media versus low media represent a physiologically relevant environment for Cryptococcus. The following concerns need to be addressed.

Major Concerns:

  1. The pH of the medias used to grow Crypto and isolate EV is not reported. Are there substantial differences in the pH of the Sabouraud Dextrose media versus the minimal media used here? Differences in pH could have a significant effect on the protein profile of EV since secretion of virulence factors are dependent on the pH of vesicles.

  1. It isn’t clear why the authors chose to examine EVs from the acap67 strain of Cryptococcus? There is not clear explanation for what role the capsule or lack of has the production of EVs or on the “type” of EVs (i.e. how would the presence or lack of capsule contribute to different protein/virulence content of EV?)

  1. The authors state on line 225, “..the sterol is the main component of the EV membrane.” What sterol are they referring to? Would this be ergosterol? Also, the authors used the Amplex Red Cholesterol assay to define the protein/sterol ratio for EVs. Can this kit evaluate ergosterol since fungal cells do not have cholesterol in their plasma membrane.

  1. If the EVs from Cryptococcus grown in minimal media showed the greatest amount of virulence factors why were mice inoculated with EVs from Cryptococcus grown in rich media (100% SDB)? Why weren’t the EVs from the minimal media examined in vivo as well?

  1. Did the authors examine the survival of animals inoculated with EVs + Cn? What happens in mice inoculated with EVs alone? Is there are difference in phenotype when mice are inoculated with EVs from rich media or minimal media?

  1. Published articles have described EVs from low carbon sources. This is physiologically relevant since low carbon mimicks the host environment. However, it is not clear how the EVs isolated here from rich media versus low media represents a physiologically relevant environment for Cryptococcus, especially since only the EVs from rich media were tested in vivo?

Minor Concerns

Line 94: The sentence should be corrected to, “ …..characteristics of the lung…”

Line 481: The sentence, “This is physiologically rationale as……” is grammatically incorrect and needs to be corrected.

Author Response

Thank you for reviewing our manuscript and suggesting corrections that will improve the overall quality of our work. Below you will find our answers regarding each suggestion and concern. Before starting the answers, I would like to talk about the importance of EVs isolated from different media. The low carbon source is described as a host tissue-mimicking condition. Considering the inflammatory status, it is true because the granulomas decrease nutrient availability. However, when the Cn infects the lung, there is an increased nutrient availability and when they migrate through the blood to other tissues and the brain. We carry out experiments to understand the fungal flexibility to release different EVs according to the tissue and increase their chances of infecting that place. Our main result is that Cn is able to sense the microenvironmental through nutritional availability and respond secreting an EV, which can establish better conditions for its growth and dissemination.

Major Concerns:

  • The pH of the medias used to grow Crypto and isolate EV is not reported. Are there substantial differences in the pH of the Sabouraud Dextrose media versus the minimal media used here? Differences in pH could have a significant effect on the protein profile of EV since secretion of virulence factors are dependent on the pH of vesicles.

Answer: The pH of the medias Sabouraud Dextrose (50% and 100%) and minimal media was all adjusted to 5.5 before experiments. It is reported now on line 110.

  1. It isn’t clear why the authors chose to examine EVs from the acap67 strain of Cryptococcus? There is not clear explanation for what role the capsule or lack of has the production of EVs or on the “type” of EVs (i.e. how would the presence or lack of capsule contribute to different protein/virulence content of EV?)

Answer: The relationship between EVs characteristics and the presence of capsule on C. neoformans was described before by Oliveira et al, 2010 (PMID: 20145096), as cited in the discussion (lines 417 to 426). The group showed that EVs from an acapsular mutant induced a higher inflammatory profile by macrophages than EVs from the encapsulated strain. It was expected, since ΔCap67 is an avirulent strain mainly due to the absence of the capsule and GXM (PMID: 6057803, PMID: 18039940), which makes it more easily recognizable by the host's immune system and eliminated. For those reasons, we wanted to search for differences in the EVs produced by both strains depending on the culture condition. This information was added now on line 102.

  1. The authors state on line 225, “..the sterol is the main component of the EV membrane.” What sterol are they referring to? Would this be ergosterol? Also, the authors used the Amplex Red Cholesterol assay to define the protein/sterol ratio for EVs. Can this kit evaluate ergosterol since fungal cells do not have cholesterol in their plasma membrane.

Answer: sterol was changed by ergosterol on line 225. We decided to use the Amplex red kit because it is a kit widely used in studies with fungi EVs to quantify sterol content (PMID: 20145096, PMID: 29795301), since this kit is sensitive to this type of molecule as well.

  1. If the EVs from Cryptococcus grown in minimal media showed the greatest amount of virulence factors why were mice inoculated with EVs from Cryptococcus grown in rich media (100% SDB)? Why weren’t the EVs from the minimal media examined in vivo as well?

Answer: The role of EVs from minimal media in lung of infected mice has been previously described (PMID: 20145096, PMID: 23144903) so treat the animals with these EVs did not bring new insights about the EVs functions. So, we decided to use the EVs produced in the rich media because when they interacted with macrophages in vitro, they induced a less inflammatory profile comparing to the EVs produced in the poor media (Figure 2) and we were interested in which would be the response in vivo when infected mice were treated with those EVs, considering that there are no studies in literature with EVs produced in a rich media, but only in a poor media. It was a confirmation about the in vitro results, showing the decrease of an inflammatory response in vivo.

  1. Did the authors examine the survival of animals inoculated with EVs + Cn? What happens in mice inoculated with EVs alone? Is there are difference in phenotype when mice are inoculated with EVs from rich media or minimal media?

Answer: We didn’t treat mice with EVs produced in the poor media. We neither inoculated mice only with EVs nor observed the survival of them because we chose to carry out the experiments that we considered more relevant to understand the inflammatory response depending on the treatment, during the initial process of infection.

  1. Published articles have described EVs from low carbon sources. This is physiologically relevant since low carbon mimicks the host environment. However, it is not clear how the EVs isolated here from rich media versus low media represents a physiologically relevant environment for Cryptococcus, especially since only the EVs from rich media were tested in vivo?

Answer: The role of EVs produced under low carbon conditions had been described before in previous studies observing macrophages and mice response to treatment with the EVs, showing a proinflammatory response (PMID: 20145096, PMID: 23144903). However, cryptococcosis is a systemic mycosis and C. neoformans is able to affect a big range of organs, migrating from the lungs, passing by blood, liver, until the brain in severe cases. These organs present different nutrient availability, requiring the fungus to adapt to them. Had been seen that the alveoli present a deficient concentration of glucose as a strategy to avoid infections, with broncho-alveolar lavage fluid containing up to 50 times less glucose than plasma (PMID: 8779986) and 10 times less than brain (PMID: 1736294). Considering the necessary to adapt to those such different conditions in host organs and that the production of EVs in a high source of carbon never had been studied, we wanted to analyze them here.

Minor Concerns

Line 94: The sentence should be corrected to, “ …..characteristics of the lung…”

            Answer: The sentence was corrected.

Line 481: The sentence, “This is physiologically rationale as……” is grammatically incorrect and needs to be corrected.

            Answer: The word “rational” was changed by “expected”.

Reviewer 2 Report

Following are the comments/suggestions for the authors to improve the current study

1) As mentioned in the methodology, the authors used GM-CSF to differentiate BMDM and BMDC. The attached cells were used as macrophages and dendritic cells from the supernatant. The research showed that the phenotypic variations can be there, and many different populations can also be isolated like basophils, eosinophils and monocytes etc. with GM-CSF differentiation (PMID: 27788572). Did authors characterize the BMDM and BMDC? Authors should provide the additional information as this can skew the findings of the study.

2) In statistical analysis authors only mentioned the analysis by ANOVA only, but in Figure 1 analysis authors used the student T-test for analysis. Authors should mention in Figure 1 panel which figures were analyzed by student T-test? In Figure 1 (E and F) authors should show the comparison with significant bars. Also, authors should provide information if SD or SEM is shown in figures.

3) The results from Figure 2A and 2B authors concluded that B3501 and ΔCap67 EVs, produced in a rich medium showed the reduction of IL-1β and TNF-? production and promoted an anti-inflammatory response by DCs. To confirm this author should check the levels of anti-inflammatory cytokines like IL-10 or TGF-beta.

4) Authors should consider shortening of introduction, should be more focused on the current research question rather than explaining the details of the various receptors involved in the immune response.

 Minor comments:

1) In Figure 4 D authors should check if 5dpi or 15 dpi? As Figure description in legend and on the figure itself is different.

2) In the methodology, authors should check for the neoformans italic font. In many places, it is not italic (eg., line 107 and 113).

3) Authors should consider expanding the abbreviations of various genes or receptors (eg., line 69, TLRs, CLRs, etc.).

4) Figure 2 quality is poor, authors should increase the size of graphs and font of the text as well.

5) From the PCR array, authors only showed the upregulated and downregulated genes. Network analysis of interacting genes can add to the manuscript and help the reader understand the key interacting genes.

Author Response

Thank you for reviewing our manuscript and suggesting corrections that will improve the overall quality of our work. Below you will find our answers regarding each suggestion and concern. Before starting the answers, I would like to talk about the importance of EVs isolated from different media. The low carbon source is described as a host tissue-mimicking condition. Considering the inflammatory status, it is true because the granulomas decrease nutrient availability. However, when the Cn infects the lung, there is an increased nutrient availability and when they migrate through the blood to other tissues and the brain. We carry out experiments to understand the fungal flexibility to release different EVs according to the tissue and increase their chances of infecting that place. Our main result is that Cn is able to sense the microenvironmental through nutritional availability and respond secreting an EV, which can establish better conditions for its growth and dissemination.

Major Concerns:

  1. As mentioned in the methodology, the authors used GM-CSF to differentiate BMDM and BMDC. The attached cells were used as macrophages and dendritic cells from the supernatant. The research showed that the phenotypic variations can be there, and many different populations can also be isolated like basophils, eosinophils and monocytes etc. with GM-CSF differentiation (PMID: 27788572). Did authors characterize the BMDM and BMDC? Authors should provide the additional information as this can skew the findings of the study.

Answer: We characterized BMDM and BMDC before the experiments by flow cytometry. We found thar BMDCs were 87% positive for CD11c and MHC class II and BMDM were 99,4% positive for CD11b, as shown in the figure below. This information was added in the text on line 157. The graphic is demonstrated in the file attached.

  1. In statistical analysis authors only mentioned the analysis by ANOVA only, but in Figure 1 analysis authors used the student T-test for analysis. Authors should mention in Figure 1 panel which figures were analyzed by student T-test? In Figure 1 (E and F) authors should show the comparison with significant bars. Also, authors should provide information if SD or SEM is shown in figures.

Answer: Information about student’s t-test and about error bars was added on methodology (line 208) and on figures’ legends and significant bars were added to Figure 1 (E and F).

  1. The results from Figure 2A and 2B authors concluded that B3501 and ΔCap67 EVs, produced in a rich medium showed the reduction of IL-1β and TNF-? production and promoted an anti-inflammatory response by DCs. To confirm this author should check the levels of anti-inflammatory cytokines like IL-10 or TGF-beta.

Answer: It was corrected on line 271, changing “promoted an anti-inflammatory response” by “didn’t promote an inflammatory response”. We didn’t measure anti-inflammatory cytokines like IL-10 or TGF-β because our first idea was to identify the influence of the EVs in the modulation of the inflammasome and their cytokines, being a virulence strategy utilized by the fungus. This idea is supported by the idea that C. neoformans activates the canonical caspase-1 and the noncanonical Caspase-8 Inflammasomes after phagocytosis, leading to a high production of pro-inflammatory cytokines (PMID: 26466953). Our group showed before that, as a strategy to avoid this response, C. neoformans produce and export small molecules that inhibit IL-1β inflammasome-dependent secretion and favors fungus to survive inside macrophage, enhancing mice infection (Bürgel et al., 2020, in press). So, our focus was the inflammasome cytokines.

  1. Authors should consider shortening of introduction, should be more focused on the current research question rather than explaining the details of the various receptors involved in the immune response.

Answer: The introduction was reformulated and is more objective.

Minor comments:

  1. In Figure 4 D authors should check if 5dpi or 15 dpi? As Figure description in legend and on the figure itself is different.

Answer: Figure description was corrected for 15 dpi.

  1. In the methodology, authors should check for the neoformans italic font. In many places, it is not italic (eg., line 107 and 113).

Answer: All C. neoformans were corrected.

  1. Authors should consider expanding the abbreviations of various genes or receptors (eg., line 69, TLRs, CLRs, etc.).

Answer: This part was removed from introduction.

  1. Figure 2 quality is poor, authors should increase the size of graphs and font of the text as well.

Answer: Figure 2 was rearranged.

  1. From the PCR array, authors only showed the upregulated and downregulated genes. Network analysis of interacting genes can add to the manuscript and help the reader understand the key interacting genes.

Answer: A network analysis of interacting genes was added in figure 4.

Round 2

Reviewer 1 Report

This authors have addressed the concerns/questions raised by previous reviewers in satisfactory manner. The manuscript is clear in its aims, results and discussion.

Author Response

Dear Reviewer, thank you for your comments. All suggestions have improved the manuscript.

Reviewer 2 Report

The revised manuscript is improved and authors have addressed the key comments and concerns. 1) Authors can include the number of technical replicates as well as the number of independent experiments performed with each figure legend.

Author Response

Thank you for the comments. We added all your suggestions in the manuscript, included a new English revision. We also have improved the conclusions. All this informations is in the manuscript version R2. 

Minor concerns:

Authors can include the number of technical replicates as well as the number of independent experiments performed with each figure legend.

This informations were adedd in all figures.